

# OF$^2$: coupling OpenFAST and OpenFOAM for high fidelity aero-hydro-servo-elastic FOWT simulations

Guillén Campaña-Alonso[1,2,*], Raquel Martín-San-Román[1,*], Beatriz Méndez-López[1], Pablo Benito-Cia[1], and José Azcona-Armendáriz[1]

[1]Wind Energy Department, Centro Nacional de Energías Renovables (CENER), Ciudad de la Innovación, 7, 31621 Sarriguren, Spain
[2]UPM, E.T.S.I. Aeronáutica y del Espacio, Universidad Politécnica de Madrid, Plaza Cardenal Cisneros, 3, 28040 Madrid, Spain
[*]These authors contributed equally to this work.

**Correspondence:** Guillén Campaña-Alonso(gcampana@cener.com)

**Abstract.** The numerical study of floating offshore wind turbines requires accurate integrated simulations, considering aerodynamics, hydrodynamics, servo and elastic response of these systems. In addition, the floating system dynamics couplings need to be included to calculate precisely the excitation over the ensemble. In this paper, a new tool has been developed coupling the NREL´s aero-servo-elastic tool OpenFAST with the Computational Fluid Dynamics (CFD) toolbox OpenFOAM. OpenFAST

is used to model the rotor aerodynamics alongside with the flexible response of the different components of the wind turbine and the controller at each time step considering the dynamic response of the platform. OpenFOAM is used to simulate the hydrodynamics and the platform's response considering the loads from the wind turbine. The whole simulation environment is called OF$^2$ (**O**pen**F**AST & **O**pen**F**OAM). The OC4 DeepCWind semi-submersible FOWT together with the NREL´s 5MW wind turbine has been simulated using OF$^2$ under two load cases. The purpose of coupling these tools to simulate FOWT is

to obtain high-fidelity results for design purposes reducing the computational time compared with the use of CFD simulations both for the rotor aerodynamics, that usually consider rigid blades, and the platform's hydrodynamics. The OF$^2$ approach allows also to include the aero-servo-elastic couplings that exist on the wind turbine alongside with the hydrodynamic system resolved by CFD. High complexity situations of floating offshore wind turbines, like storms, yaw drifts, weather-vane, or mooring line breaks, that implies high displacements and rotations of the floating platform or relevant non-linear effects can

be resolved using OF$^2$, overcoming the limitation of many state of the art potential hydrodynamic codes that assume small displacements of the platform. In addition, all the necessary information for the FOWT calculation and design processes can be obtained simultaneously, such as the pressure distribution at the platform components and the loads at the tower base, fairleads tension, etc. Moreover, the effect of turbulent winds and/or elastic blades could be taken in account to resolve load cases from the design and certification standards.





## 1 Introduction

Floating offshore wind turbines (FOWT) design and optimization is necessary to accomplish the requirements with regard to the increase of wind energy capacity installed worldwide. The reduction of the LCOE of offshore wind energy will be possible, among others, if the fidelity of the tools used to design FOWT is improved without a great increase of computational time. In addition, the coupling of the wind turbine and platform dynamics is necessary to the ensemble optimizations necessary in wind turbine and platform co-design processes.

Most of the state of the art hydrodynamic models used in engineering simulation tools, for the coupled analysis of floating offshore wind turbines (FOWT), are based in two different hydrodynamic models to resolve the hydrodynamic loads on the floating platform: Morison's equation (ME) and potential flow theory (PF). The ME (see Morison et al. (1950)) can be applied to slender bodies and provides the inertia and drag forces over these elements. The PF (see Newman, J.N. (1977); Faltinsen, O.M. (1993)) is applicable to general geometries to solve the hydrodynamic problem, obtaining the added mass, radiation damping, diffraction forces, etc., but does not include viscous effects. The viscous effects can be added to potential models through the drag term of Morison's equation, or by adjusting the damping of the platform based in experimental data (see Azcona (2016)), or Computational Fluid Dynamic (CFD) simulations. This potential solution can be obtained both in the frequency and time domains. Moreover, the forces and moments obtained by solving the potential problem in the frequency domain can be introduced into a time domain solver of the floating platform, see for example Jonkman, J.M. (2007).

As mentioned before, the hydrodynamic response of floating platforms can also be modelled performing high fidelity CFD simulations. This method has became, nowadays, part of the design process of FOWT. These simulations support the design process and allow tuning the integrated numerical tools since the early stages of the process, so that the effort in wave tank testing can be kept once a mature platform design has been achieved. CFD simulations are used to provide quantitative information to the design process such as the damping coefficients needed in the engineering codes. In addition, flow phenomena such as wave run up or pressures over the structure, or the heave plates, are provided to optimize the platform design and to understand its dynamics. Several publications can be found in which CFD is applied to simulate platform hydrodynamics. For instance, the OC6 Phase I collaborative work under the IEA Task 30 provided two publications, in the first one the platform response to bi-chromatic waves was analysed in Wang et al. (2021), making special focus in the waves treatment, pressures over the structure and wave run-up analysis. In the second one, free decay simulations were performed to make a benchmark between different CFD codes, including a detailed comparison with experiments described in Wang et al. (2022). Both publications demonstrated the potential of CFD use in platform design and characterization, and pointed out the difference with regard to potential-flow solvers simulations. For example, it has been found that the potential-flow solution used in Wang et al. (2021) significantly under-predicts the damping of surge motion.

On the other hand, rotor aerodynamics are simulated in the wind energy industry with different fidelity level tools ranging from blade element momentum theory (BEMT) Bossanyi et al. (2001); Bladed (2010), more complex free vortex filament



methods (FVM) Kecskemety and McNamara (2011); Marten et al. (2019), actuator line approaches Quon et al. (2019); Bran-
lard et al. (2014), and the high fidelity fully-resolved CFD simulations. Typically, BEMT and FVM approaches are used for
coupled aeroelastic simulations, while the different CFD approaches are used in purely aerodynamic simulations without con-
sidering the coupling with flexible degrees of freedom. Moreover, CFD is mainly used in the airfoil level or to specific cases in
which extreme aerodynamic events need to be deeply analysed. Recently, in the OC6 Phase III project numerous aerodynamic
models with different fidelity levels have been compared, in purely aerodynamic conditions, against wind tunnel experimental
data of a wind turbine placed over a moving structure capable of imposing displacements and rotations on the tower base of
the wind turbine Bergua et al. (2022). This study has shown that all analyzed aerodynamic models are capable of accurately
predict the aerodynamic loads under the forced pitch and surge motion studied in this OC6-Phase III project. However, it has
been found that when considering the additional dynamics introduced by the controller the aerodynamic cycles change.

Furthermore, the combined hydro-aero high fidelity simulations of FOWT under wind and wave conditions is a cutting edge
technology with few research works available in the literature Otter et al. (2021); Micallef and Rezaeiha (2021). In addition,
in the few existing models it is very rare to see couplings with elastic models of the flexible elements of the wind turbine, such
as the blades or the tower. And it is even more difficult to find models that include the coupling with the wind turbine control
system. Ren et al. (2014) made a CFD analysis of the NREL 5-MW with a TLP structure under wind and wave conditions and
simulated with the commercial software FLUENT. In that work only the surge motion was allowed. Liu et al. (2017) presented
in their work a coupled CFD simulation using OpenFOAM both in the rotor and in the floating platform. No information was
provided about the computational time of that simulations. Tran and Kim (2016) carried out fully coupled aero-hydrodynamic
simulations of the OC4-DeepCWind semi-submersible with a wind turbine using CFD and a catenary mooring solver. The
major FOWT components were simulated without considering structure deformations. The results considering free decay tests
and regular wave conditions showed good agreement with the MARIN tests and the FAST code. Zhang and Kim (2018) also
carried out a fully coupled aero-hydrodynamic simulations of the DeepCwid semi-submersible with the NREL 5-MW wind
turbine and also compared with experimental measurements of the OC5 project Robertson et al. (2017). In this work, the sim-
ulation time for one case was 20 days with 66 CPUs. In addition, it was found that the power output is more sensitive than the
thrust force to platform motions.

Moreover, in the design and certification process of FOWT, following standards such as IEC-61400-3-2 Ed1 International
Electrotechnical Commission (2019) or NI572 Bureau Veritas (2019), the hydrodynamic pressure over the surface of the plat-
form may be requested alongside with the loads at tower base or mooring tensions at the fairleds for different cases with the
wind turbine in normal operational state, storms or under fault conditions. Even more, some specific FOWT designs equipped
with single point mooring (SPM) may have large rotations in order to weather-vane with the wind, that can violate some limi-
tations or assumptions of the state of the art design codes like OpenFAST (see Jonkman (2009)). Therefore, a new simulation
tool is presented in this work, called OF$^2$, that combine a high fidelity representation of the hydrodynamic behaviour of the
floating platform with an aero-servo-elastic representation of the tower and rotor-nacelle assembly. This approach reduces the



computational time with regard to full CFD simulations of FOWT, allowing to introduce the control system in the simulation
and including flexible response of the different FOWT components. The dynamic pressure, the mooring tension, wave run-up
and the body forces can be obtained as in the visualization example that has been represented in Fig. 1.

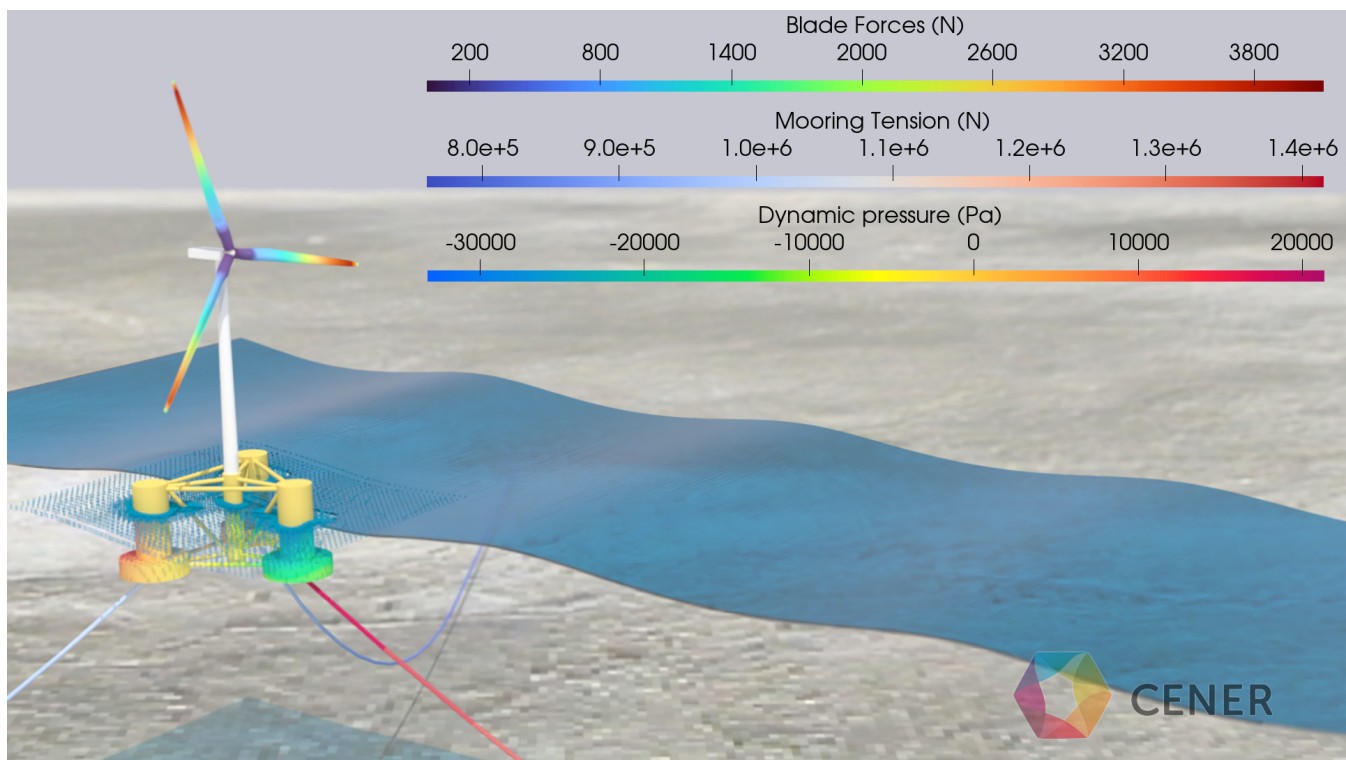

**Figure 1.** Visualization of an OF$^2$ simulation. The forces on the blades are shown alongside the mooring line tension and the dynamic
pressure on the platform.

The rest of the article is organized as follows: the methodology to couple OpenFAST and OpenFOAM is defined in Sect.
2, then the verification methodology is included in Sect. 3. It includes, firstly, the description of the load cases simulated
to demonstrate the applicability of the method and the advantages with regard to potential codes or fully CFD simulations.
Secondly, the FOWT model used to test OF$^2$ will be described as well as the simulations set-up and the results. Finally, the
conclusions of this work will be presented in Sect. 4.

## 2 OF$^2$ methodology: OpenFAST and OpenFOAM coupling

In this work OpenFAST and OpenFOAM are coupled in order to better simulate the floating platform's hydrodynamic response
and to overcome engineering models limitations. With the following approach, the aero-servo-elastic response of the wind tur-





bine is simulated with OpenFAST, while the floating platform dynamics and fluid flow are simulated with OpenFOAM. The resulting tool has been named OF$^2$.

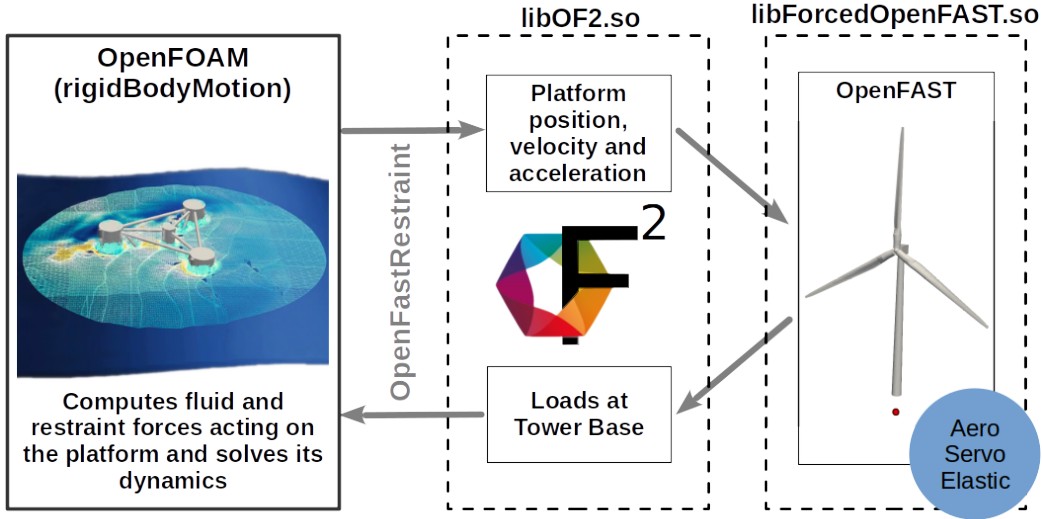

**Figure 2.** Flowchart of OF$^2$ coupling process.

Hence, this OF$^2$ environment has been made through the development of two new shared libraries. The operation scheme

of all the OF$^2$ libraries within OpenFOAM can be seen in Fig. 2. Firstly, `libForcedOpenFAST.so` has been developed. This library allows to run OpenFAST imposing the floating platform displacements (see Martín-San-Román (2022) for details of imposition of movements in OpenFAST). Secondly, a new Rigid Body Motion type restraint, named `libOF2.so`, has also been created. This `libOF2.so` restraint uses the functions existing inside `libForcedOpenFAST.so` in order to apply the loads computed by OpenFAST on the Rigid Body, i.e., the floating platform. Therefore, at each time step, the floating platform

dynamics is solved by the Rigid Body Motion library within OpenFOAM. When the OF$^2$ restraint is executed, it uses the displacement, velocity and acceleration of the floating platform as an input for the functions of `libForcedOpenFAST.so` that impose this displacement to the wind turbine modelled within OpenFAST and calculate the corresponding loads, power and deformations of the different wind turbine components. Finally, the loads at the tower base point are then applied to Open-FOAM's body, along with the ones resulting from the other restraints (like mooring lines or external forces if any) and fluid

forces. Once the platform´s dynamics response is solved, the mesh is updated and adapted to the new platform´s position and the fluid flow is solved finishing the current time step iteration. This approach ensures that the effect of the platform dynamics over the tower and rotor nacelle assembly is considered in both the servo, elastic and aerodynamic response of each of these components and vice versa. An example of a simplified dynamicMeshDict file used in OpenFOAM to describe the body dy-





namics using the new shared libraries can be seen in Listing 1.

```
 1: /*--------------------------------*- C++ -*----------------------------------*\
 2: | =========                 |                                                 |
 3: | \\      /  F ield         | OpenFOAM: The Open Source CFD Toolbox           |
 4: | \\    /   O peration      | Version:  v2106                                 |
 5: | \\  /    A nd            | Web:      www.OpenFOAM.com                      |
 6: | \\/     M anipulation    |                                                 |
 7: \*---------------------------------------------------------------------------*/
 8: FoamFile
 9: {
10:     version     2.0;
11:     format      ascii;
12:     class       dictionary;
13:     object      dynamicMeshDict;
14: }
15: // * * * * * * * * * * * * * * * * * * * * * * * * * * * * * * * * * * * * //
16: dynamicFvMesh       dynamicMotionSolverFvMesh;
17: motionSolver        rigidBodyMotion;
18: motionSolverLibs
19: (
20:     "librigidBodyMeshMotion.so"
21:     "libmoordynRestraint.so"
22:     "libOF2.so"
23: );
24: rigidBodyMotionCoeffs
25: {
26:     ...
27:     bodies
28:     {
29:         platformBody
30:         {
31:             type            rigidBody;
32:             parent          root;
33:             ...
34:         }
35:     }
36:     restraints
37:     {
38:         OpenFastRestraint
39:         {
40:             type                    OpenFast;
41:             body                    platformBody;
42:             openfast_file           "path/to/fst/file";
43:             initial_rotation        (x y z);
44:             initial_position        (x y z);
45:             fromJtoLoadApplicationPoint (x y z);
46:             fromJtoPtfmReferencePoint   (x y z);
47:         }
48:         MoordynRestraint
49:         {
50:             type                    moordyn;
51:             body                    platformBody;
52:             fromJtoPtfmReferencePoint  (x y z);
53:         }
54:     }
55: }
```

**Listing 1.** Extract of the dynamicMeshDict file where it can be seen the usage of the new restraints, libOF2.so and libmoordynRestraint.so.



# 3 Verification of the methodology

## 3.1 Load Cases

In order to verify OF$^2$, two verification load cases have been evaluated with OF$^2$ and an OpenFAST-only approaches. The two cases have been based on the Load Case (LC) 3.1 of the OC4 project Robertson et al. (2014b), with a steady uniform wind speed of 8 m/s and a regular wave height ($H$) of 6 m and a period ($T$) of 10 s. In the first load case analyzed in this work, called 3.1*, no waves have been included. All the main characteristics of these two load cases have been summarized in Table 1.

**Table 1.** Description of the Load cases analysed, adapted from OC4 Phase II Robertson et al. (2014b).

| Load Case | 3.1* | 3.1 |
|---|---|---|
| Description | Deterministic at below rated | Deterministic at below rated |
| Wind turbine initial condition | $\Omega = 9$ rpm<br>blade pitch = 0 degrees<br>nacelle yaw = 0 degrees | $\Omega = 9$ rpm<br>blade pitch = 0 degrees<br>nacelle yaw = 0 degrees |
| Enabled DOFs | All | All |
| Wind Condition | Steady, uniform, no shear<br>$Vhub$ = 8 m/s | Steady, uniform, no shear<br>$Vhub$ = 8 m/s |
| Wave Condition | No wave | Regular Stokes II:<br>$H$ = 6 m,<br>$T$ = 10 s |

## 3.2 Simulation set-up

The new tool, OF$^2$, has been used to evaluate the response under wind and waves loading. For this study, OpenFAST v2.6.0 and OpenFOAM v21.06 have been coupled to model the NREL 5-MW wind turbine on the OC4 semi-submersible DeepCWind floating platform (see Jonkman et al. (2007) and Robertson et al. (2014a)).

The tower and rotor nacelle assembly have been modelled considering the flexibility of the different components. For the three blades, two flexible modes in flap-wise direction and one in edge-wise direction have been considered. Additionally, for 190 the drive-train, a torsional mode has been included and two flexible modes have been also considered, both in fore-aft direction and side-side direction, to represent the tower flexible response. The floating platform is considered as a fully rigid structure. Furthermore, an in-house controller designed for this FOWT has been used.

Moreover, the mooring system has been simulated using MoorDyn (see Hall (2017)) using the OpenFOAM's restraint developed by Chen and Hall (2022). This restraint has been modified to work together with the OpenFOAM's Rigid Body Motion





library and it has been called `libmoordynRestraint.so`. The way to include this new restraint in the dynamicMeshDict, is also included in Listing 1.

For the CFD simulations performed inside OF[2] an unstructured mesh has been created with snappyHexMesh, where the domain size is 581m/403m/278m in the surge, sway and heave directions. The smaller element on the platform's surface mesh

has a size between 0.3 and 0.6 m. Different mesh details are shown in Fig. 3 (overall view), Fig. 4 (platform body view) and Fig.5 (platform surroundings view).

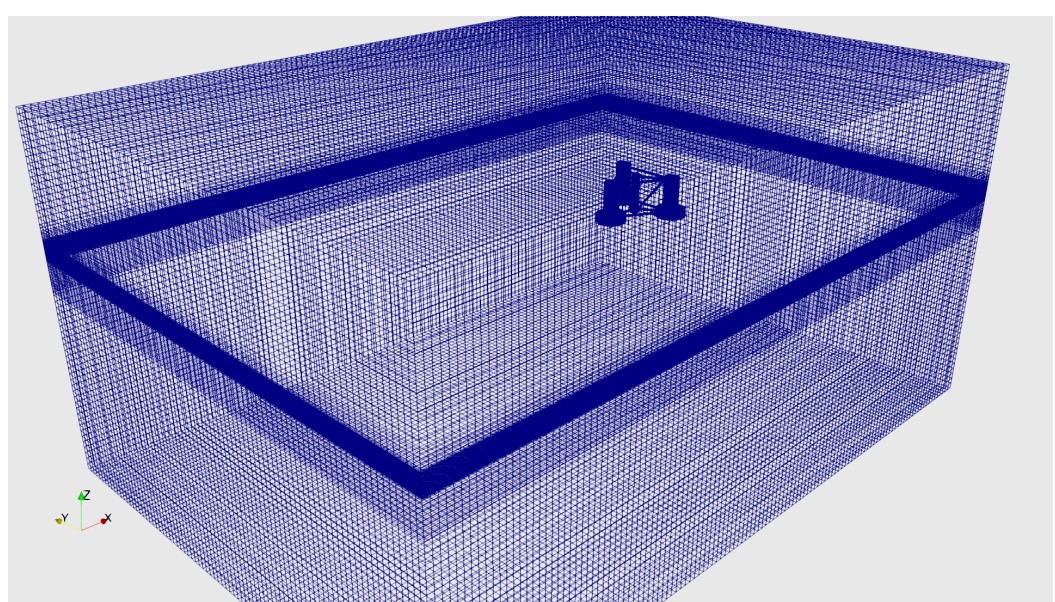

**Figure 3.** Computational domain mesh, overall view.

Regarding the numerical schemes used, first order implicit laminar simulations with OpenFOAM v21.06 have been done. In particular, Gauss linear spatial schemes, for the gradient terms, and Gauss upwind and Gauss MUSCL schemes, for the

205 divergence terms, have been used. Also, MULES interface capturing scheme has been selected. Finally, the PIMPLE algorithm has been used to solve the pressure-velocity coupling. The under-relaxation factors for both velocity and pressure have been set to 1. As the simulation of a floating platform movement needs from a dynamic mesh approach, a morphing mesh technique has been selected to be used in this work to accommodate the motion of the floater. Additionally, the displacement Laplacian, as the motion solver, and the moving wall, as the boundary condition in the floating platform, have been used. With an im-

210 plicit algorithm the mesh morphing is updated at each iteration driven by the platform dynamics. Finally, the used boundary conditions are wave velocity inlet and pressure outlet in the inlet and outlet boundaries, the ground is considered as a wall and the domain sides are modelled with an slip condition. Moreover, the boundary condition used for wave generation uses a ramp





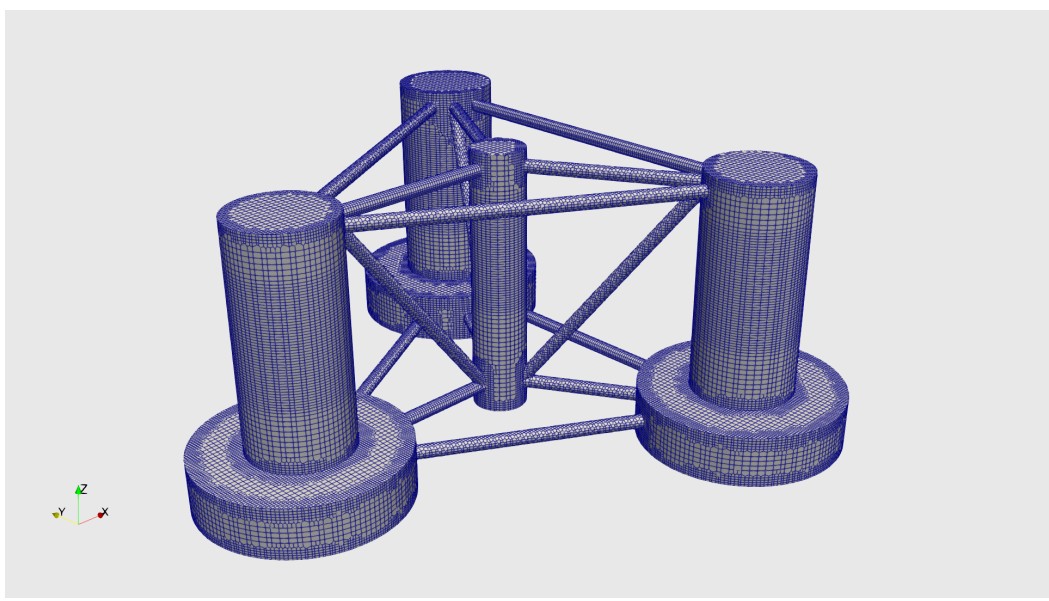

**Figure 4.** Platform surface mesh.

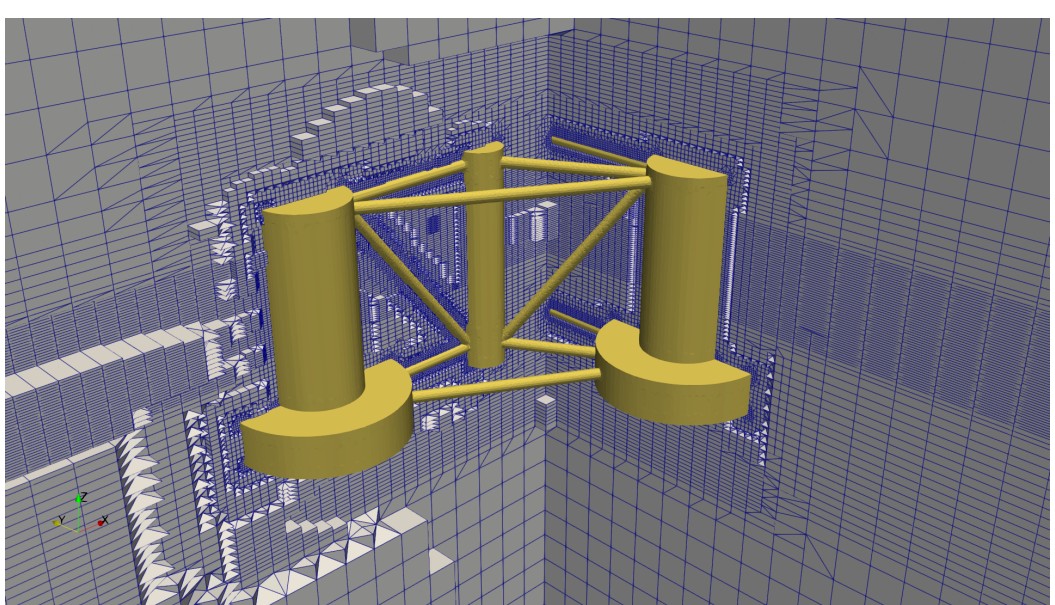

**Figure 5.** Near platform refinement.



time scale factor to avoid numerical divergence. The resulting wave elevation profile has an initial transitory state were the wave amplitude is gradually increased. This transient evolution is shown in Fig. 6.

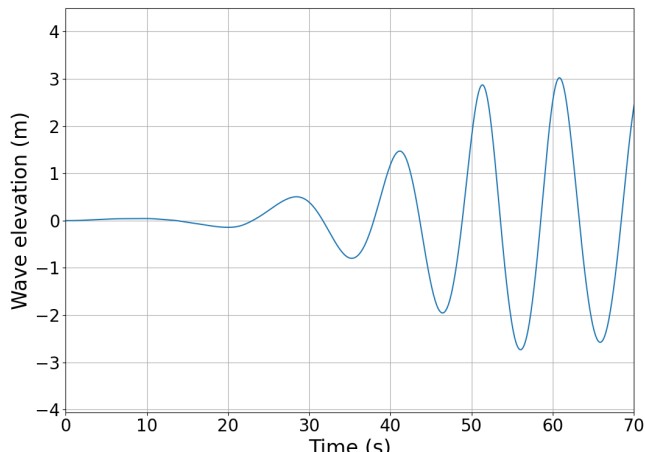

**Figure 6.** Wave elevation transient evolution.

In order to analyse the OF$^2$ performance an OpenFAST-only model for comparison purposes has been created to define the complete integrated model of the FOWT. It has to be noted that the OpenFAST model for the tower and RNA (Rotor Nacelle Assembly) is the same in the OpenFAST-only model and the coupled tool OF$^2$. In particular, the same BEM (Blade Element Model) approach has been applied to compute the aerodynamic loads at the rotor, and the same wind files have been used in both simulations. The ElastoDyn representation of the tower and rotor nacelle assembly in OpenFAST is the same used in the OF$^2$ solver as well as the same MoorDyn input files. The wave elevation profile used in OpenFAST to generate the wave kinematics has been extracted from OF$^2$ simulations of the undisturbed wave train. It should be noted that the wave elevation signal used in OpenFAST simulation of LC 3.1 has been extracted from an empty channel simulation performed with OpenFOAM, this is, from a simulation of the sea state without the floating platform. Therefore, the waves that affect the dynamics of both OpenFAST and OF$^2$ simulations are exactly the same. In the OpenFAST-only simulations, the platform's hydrodynamic response has been represented through the HydroDyn module (see Jonkman (2009)) with a combination of potential-flow and Morison equation. The drag coefficient of the members range between 0.56 and 0.68, depending on the diameter, as is defined in Robertson et al. (2017). A drag coefficient of 9.6 has been used for the heave plates, using the plates area as reference to compute the force. Non-linear hydrodynamics has been included using full QTF (Quadratic Transfer Functions).The two load cases computed in this work have a simulation time of $400$ s with a time step of $0.01$ s. The OF$^2$ simulations have been run on 1 node each equipped with dual AMD EPYC 7543 32-core processor and 128 GB of RAM.





## 3.3 Results

Hereafter, the results obtained by both approaches, OpenFAST-only and OF$^2$ are compared. Firstly, the time series results of the platform's degrees of freedom (DOF) are compared in order to have a qualitative comparison of the results obtained from both OF$^2$ and OpenFAST tools. Then, a quantitative comparison of the mean and standard deviation values of these DOF have also been performed.

The main platform's DOF results can be seen in Fig. 7. This figure includes the results obtained for surge (top), heave (middle) and pitch (bottom). The sub-figures of the left column represent the results of LC 3.1* (i.e., the case with still water), while the sub-figures on the right column represent the results for LC 3.1 (i.e., including the regular wave).

As it can be seen in Fig. 7, both simulation approaches present similar platform's dynamic responses. In particular, for the case in still water, the signals of the transient caused by the constant wind loading are very close in both models for the surge and pitch motions. A slight difference in the mean heave position of around 0.05 m between both approaches may be caused by small differences in the displaced volume. Once the regular wave is introduced, more differences between both simulation approaches arise. In particular, the surge motion predicted by both codes in Fig. 7b, present a different transient evolution. Nevertheless, once both simulations reach the stationary position, it can be observed that the amplitude of the motion is very similar. The mean displacement is also equivalent in both simulations, showing that mean drift forces from non-linear effects are similar. In Fig. 7c, a small difference in the mean heave position is appreciated. This is consistent with the difference observed for the still water case. In addition, the amplitude of the heave motion for the OpenFAST-only simulation is larger than for OF$^2$. Similarly, OpenFAST-only approach also predicts a large amplitude in the pitch motion compared to OF$^2$ (see Fig. 7f). These differences in the amplitude for the heave and pitch motions could be caused by the simplified treatment of the viscous loads done in the OpenFAST, in particular for the heave plates.

The force results shown in Fig. 8, follow the same scheme as the motions in Fig. 7. In particular, Fig. 8 compares the resulting hydrodynamic forces acting on the platform for each modelling approach. Moments are computed with regard to the platform reference point. The loads computed under the OF$^2$ approach are those exerted by the fluid on the platform, i.e., both hydrodynamic and hydrostatic loads. Whereas, the demanded loads output under the OpenFAST approach are the integrated hydrodynamic loads. Therefore, it must be noted that these OpenFAST's loads also take into account hydrostatic forces. Firstly, it must be noted that both approaches determine similar mean loads. For the case with still water, the surge force and pitch moment are very similar. The small scale differences in the heave force amplitudes are caused by the larger motions of the OF$^2$ simulation, that is initialized at farther position from its equilibrium, compared to the OpenFAST-only simulation. However, once the regular wave is introduced, the amplitude of the forces is higher in the OpenFAST-only approach, specially for the heave and pitch degrees of freedom. This is consistent with the results for the motions already presented in Fig.7, and it is likely that these differences are caused by the different hydrodynamic approaches, in particular for the viscous effects.





**Figure 7.** Platform response in surge (top), heave (middle), pitch (bottom) degrees of freedom in still water case, LC 3.1*, (left column) and regular wave case, LC 3.1, (right column). The results obtained with OpenFAST have been presented in blue while $OF^2$ have been represented with orange.



(a) LC 3.1* Surge force

(b) LC 3.1 Surge force

(c) LC 3.1* Heave force

(d) LC 3.1 Heave force

(e) LC 3.1* Pitch moment

(f) LC 3.1 Pitch moment

**Figure 8.** Hydrodynamic forces in surge (top), heave (middle), pitch (bottom) directions in still water case, LC 3.1*, (left column) and regular wave case, LC 3.1, (right column). The results obtained with OpenFAST have been presented in blue while OF$^2$ have been represented with orange.





In addition,since the $OF^2$ approach solves the fluid domain, the pressure distribution on the platform surface, among other outputs, is available for further analysis reinforcing the suitability of this tool for co-design processes. For example, in Fig. 9, the dynamic pressure distribution over the floating platform is shown at a particular instant of the simulation. Additionally to the high fidelity simulation of the platform dynamics, with $OF^2$ it is also possible to include the control and flexibility response of the wind turbine with a lower computational effort than with a fully flexible CFD approach. Therefore, the flexible response

predicted by $OF^2$ at the tower top and the blade tip locations have been compared against OpenFAST-only simulations in Fig. 10 and Fig. 11, respectively, only for LC 3.1 with regular wave. The comparison of these variables for the still water case are not included for simplicity. Nevertheless, the comparison for that case is equivalent. In these figures it can be seen that the differences in amplitude, specially for the pitch platform rotation that have been previously observed in Fig. 7f, are also visible in the tower top fore-aft displacement and blade tip out-of-plane deflection in Fig. 10a and 11a, respectively.

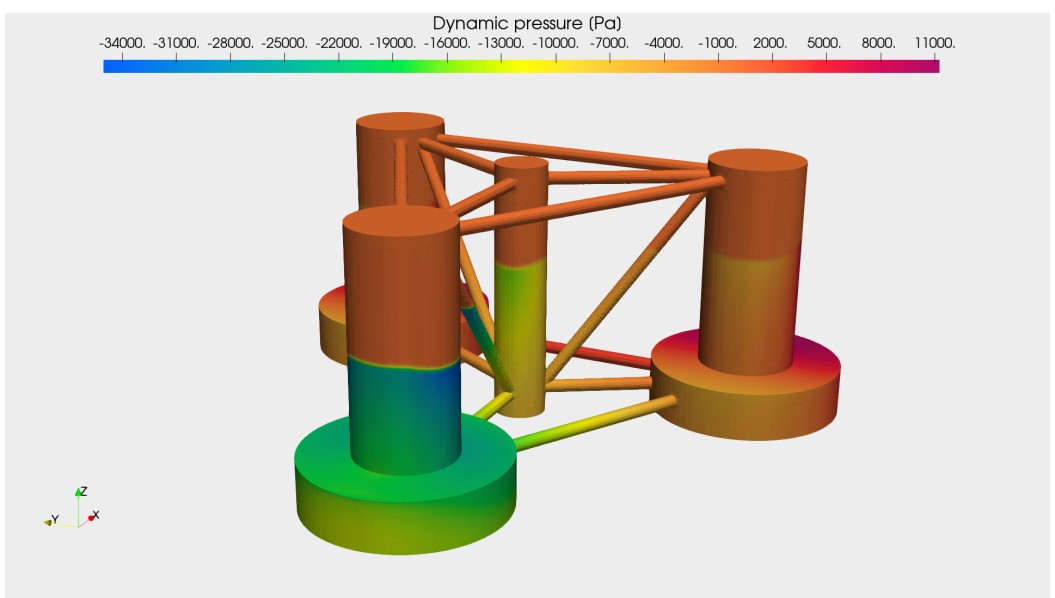

**Figure 9.** Pressure distribution over the floating platform at time 300 s.

Moreover, the control performance for the regular wave case LC 3.1, is presented in Fig. 12. Both, rotational speed (Fig. 12a) and generator power (Fig. 12b) present a slightly lower mean value in $OF^2$ than in OpenFAST-only and a smaller amplitude.

Finally, the statistical analysis of all these time signals has been included in Table 2. In this table, the standard deviation (std) and the mean values (mean) for the two approaches compared in this work have been included. Additionally, the differences

obtained between the two models have been quantified in terms of normal differences as shown in Eq. 1:

$$Diff[\%] = 100\frac{OF^2 - OpenFAST}{OpenFAST} \tag{1}$$



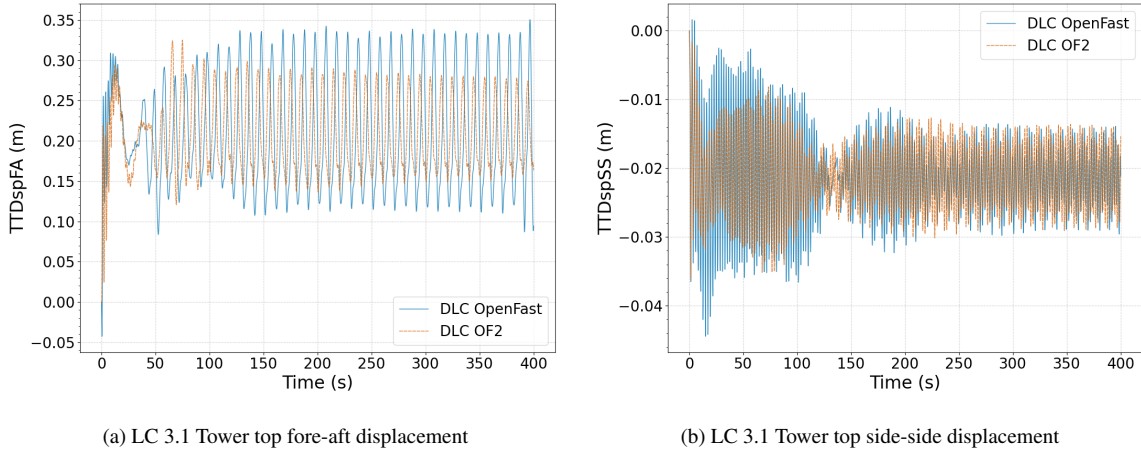

(a) LC 3.1 Tower top fore-aft displacement

(b) LC 3.1 Tower top side-side displacement

**Figure 10.** Tower top deformations for the regular wave case, LC 3.1.Tower top fore-aft deflection (left) and tower top side-side deflection (right). The results obtained with OpenFAST have been presented in blue while OF$^2$ have been represented in orange.

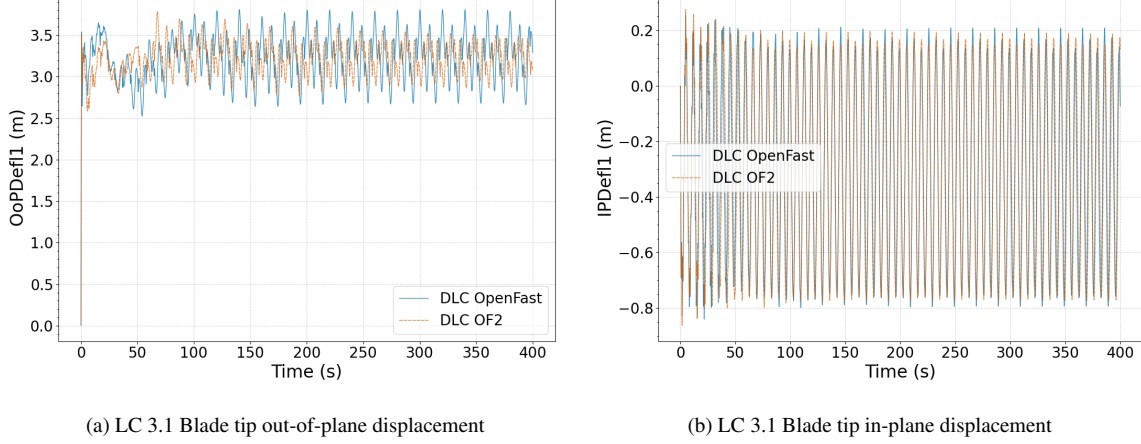

(a) LC 3.1 Blade tip out-of-plane displacement

(b) LC 3.1 Blade tip in-plane displacement

**Figure 11.** Blade tip deformations for the regular wave case, LC 3.1. Blade tip out-of-plane deflection (left) and blade tip in-plane deflection (right). The results obtained with OpenFAST have been presented in blue while OF$^2$ have been represented in orange.



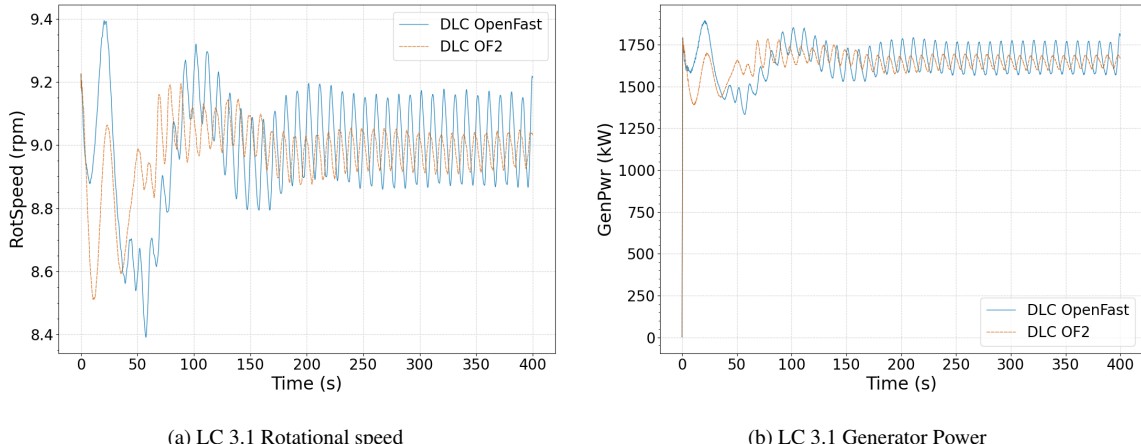

(a) LC 3.1 Rotational speed          (b) LC 3.1 Generator Power

**Figure 12.** General regulation variables for the regular wave case, LC 3.1. The rotational speed (left) and the generator power (right). The results obtained with OpenFAST have been presented in blue while OF$^2$ have been represented in orange.

Therefore, with this metric, if the difference is a positive value means a higher value in OF$^2$ than in OpenFAST-only results. This metric has been applied for both the standard deviation and the mean value. It is noticeable in Table 2 that the higher differences between OF$^2$ and OpenFAST-only approaches are obtained in platform sway and heave DOF. However, as these degrees of freedom have a very small range, it must be stated that the actual difference (without normalizing) is less than 10 mm in sway and 12 cm in heave.

The approach proposed in this work, using OF$^2$ to perform coupled simulations of floating offshore wind turbines, present advantages both over the lower fidelity resolution and over other high fidelity approaches found in the literature. For example, when comparing OF$^2$ capabilities with potential flow hydrodynamic solvers, OF$^2$ allows to include higher order terms and viscous effects that are more difficult to fit in lower fidelity models like HydroDyn. Moreover, OF$^2$ will allow to overcome the limitation of HydroDyn that assumes small rotations for the platform response applying the hydrodynamic loads without updating these rotations and taking into account the actual position of the free surface. This advantage makes OF$^2$ a recommendable tool for detailed analysis of the response of concepts equipped with SPM since they do not have any restrictions for rotation around the vertical axis. Additionally, OF$^2$ present lower computational costs than others fully coupled high fidelity simulations found in the literature. To quantify this difference in computational cost Table 3 has been included. This table specifies for each tool used in this study and those from Tran and Kim (2016) and Zhang and Kim (2018), some details of the modelling methodology, the number of cores used for the simulation, the simulated time, and the time it took to complete the simulation. As it can be seen, OF$^2$ has a much higher computational cost than OpenFAST-only approach. However, it still allows ten-minute load simulations to be carried out in less than 1 day. Moreover, with OF$^2$, detailed simulations of complex



**Table 2.** Statistical results of the different variables analyzed for the load case under wind and waves LC 3.1. The standard deviation (std) and the mean values (mean) for each model used, OpenFAST and OF$^2$ have been included, alongside with the normalized differences obtained between the two models following Eq. 1

| Variable | Units | OpenFAST | | OF$^2$ | | Diff [%] | |
|---|---|---|---|---|---|---|---|
| | | std | mean | std | mean | std | mean |
| Platform Surge | (m) | 3.36 | 4.62 | 1.76 | 4.55 | -47.53 | -1.60 |
| Platform Sway | (m) | 0.01 | 0.00 | 0.01 | 0.01 | 94.45 | 2134.41 |
| Platform Heave | (m) | 0.55 | 0.06 | 0.41 | -0.06 | -25.79 | -204.73 |
| Platform Roll | (deg) | 0.02 | 0.10 | 0.01 | 0.10 | -6.57 | -0.84 |
| Platform Pitch | (deg) | 0.93 | 1.89 | 0.79 | 1.79 | -15.22 | -5.03 |
| Platform Yaw | (deg) | 0.03 | -0.06 | 0.02 | -0.06 | -31.26 | 6.4 |
| Blade tip In-plane Displacement | (m) | 0.33 | -0.30 | 0.33 | -0.30 | -0.24 | 0.05 |
| Blade tip Out-of-plane Displacement | (m) | 0.29 | 3.23 | 0.21 | 3.22 | -27.15 | -0.09 |
| Tower Top Fore-aft Displacement | (m) | 0.07 | 0.21 | 0.05 | 0.21 | -31.86 | -1.57 |
| Tower Top Side-side Displacement | (m) | 0.01 | -0.02 | 0.01 | -0.02 | -21.07 | -0.35 |
| Generator Power | (kW) | 107.88 | 1649.92 | 76.29 | 1641.35 | -31.14 | -0.52 |
| Rotational Speed | (rpm) | 0.17 | 8.99 | 0.11 | 8.98 | -35.75 | -0.15 |

cases can be addressed using less than 5% of the computational resources necessary for a complete CFD approach for both aero and hydro dynamics.

**Table 3.** Computational cost of different tools used for the coupled analysis of FOWT under wind and wave loading

| Tool | Hydrodynamic | Aerodynamic | Flexibility | Controller | Simulated time | Cores | Wall-clock time | Core hours |
|---|---|---|---|---|---|---|---|---|
| OpenFAST | PF and ME | BEMT | Yes | Yes | 400 s | 1 | 7 minutes | 0.1167 |
| OF$^2$ | CFD-URANS | BEMT | Yes | Yes | 400 s | 64 | 13.1 hours | 838.4 |
| Tran and Kim (2016) | CFD-URANS | CFD-URANS | No | No | 500 s | 32 | 24 days | 18432 |
| Zhang and Kim (2018) | CFD-URANS | CFD-URANS | No | No | 300 s | 66 | 20 days | 31680 |

## 4 Conclusions

A new simulation tool, called OF$^2$, for time domain simulations of FOWT has been developed. The main conclusions of this
work can be summarized as follows:

    – OF$^2$ combines a high fidelity resolution of the hydrodynamic response of a floating platform with a multi-complexity
      aero-servo-elastic tool for the simulation of the wind turbine.





- With the coupling of OpenFAST to a CFD simulation of the platform hydrodynamics, all the potential from OpenFAST can be used to introduce the wind turbine components flexible behaviour, turbulent winds and the control laws necessary for the FOWT operation.

- The new tool has the advantage of a reduction of 95% of the computational time with regard to the use of a full CFD approach that includes the turbine aerodynamics.

- Load cases with big platform displacements and wind turbine operation events can be simulated with $OF^2$. Current engineering tools present limitations in accurately capture the effect of large displacements and state of the art CFD simulations typically consider rigid rotors.

- $OF^2$ has been verified in this study against OpenFAST-only simulations. The OC4 semi-submersible floating platform Robertson et al. (2014a) and the NREL 5 MW wind turbine Jonkman et al. (2007), under co-directional wind and wave loading, has been used in this verification. The results have shown that the principal platform degrees of freedom present very similar mean values between the $OF^2$ and the OpenFAST-only approaches, in particular for the wind-only cases. Once the regular waves are introduced, higher differences arise, specially for the heave and pitch motions. It is likely that this is caused by the different treatment of the viscous forces, that $OF^2$ solves using a higher complexity approach.

- In addition, as $OF^2$ solves the complete fluid domain, it provides a detailed representation of the distributed magnitudes on the platform surface, which can be useful for the calculation and design process. For example, it can be obtained simultaneously the pressure distribution at platform components and the loads from the tower, the anchoring system, etc.

- $OF^2$ could be used as part of the FOWT co-design techniques to optimize the design and therefore, contribute to the reduction of LCOE of offshore wind energy.

- With $OF^2$, an advance in the state of the art of simulation codes for FOWTs has been done. This will support the reduction of offshore wind energy cost reduction needed to boost the maturity of floating offshore wind energy.

In future works $OF^2$ will be used to analyze SPM designs to study weather-vaning response under co-directional and misaligned wind and wave loading. Moreover, $OF^2$ will be used to obtain the required distributed loads over the platform surface, alongside with the loads from the fairleads and tower base, to be used in an structural simulation tool for the analysis of ultimate and fatigue loads over the floating structure. $OF^2$ will be also used coupled with MUST Martín-San-Román (2022), an in-house tool, based in OpenFAST, for the coupled analysis of multi wind turbine floating platforms. This will allow analyzing the response of these type of configurations when equipped with SPM. MUST includes a free vortex filament method (FVM) module for the rotor aerodynamics, that will provide more accurate prediction of aerodynamic loads in the misaligned conditions that arise under large displacements of the system.

*Author contributions.* Guillén Capaña-Alonso: Tool development, methodology definition, verification, $OF^2$ simulations and writing. Raquel Martín-San-Román: Tool development, verification, methodology definition, OpenFAST simulations and writing. Pablo Benito-Cia: Tool




development, methodology definition and verification. Beatriz Méndez-López: Funding acquisition, conceptual definition and writing, José

Azcona-Armendáriz: Results analysis, verification and writing.

*Competing interests.* The authors declare that they have no conflict of interest.

*Acknowledgements.* This work has been funded by the Government of Navarra under the scope of the COSTA project (grant number 0011-1383-2022-000000 )



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
