# Peer review of "OF2: coupling OpenFAST and OpenFOAM for high fidelity aero-hydro-servo-elastic FOWT simulations"

_Wind Energy Science, 2023_

## Referee Comment (RC2)

This paper develop a new tool coupling OpenFAST and OpenFOAM where the hydyodynamics and dynamic reponses of platfrom is conisdered in OpenFOAM and the aerodynamics of blade and controller in simulated in OpenFAST, to obtain the high-fidelity results, especially the nonlinear hydrodynamic loads. I believe this paper more or less makes contribution to the community. However, there are some technical issues must be addressed to increase the quality of the paper. Pleases address them or rebut them in your answer to present review:

Major comment:

As the hydrodynamics of platform is considered in OpenFOAM, and the motions of platform is transmitted to OpenFAST. Why the OpenFAST calculate the hydrodynamics again based on potential-flow and Morison equation? I think the coupling between OpenFOAM and OpenFAST should be explained in detail.

Minor comments:
1. Listing 1 can be given as an appendix.
2. I think it needs some words to introduce how the mesh around the platform's surface is refined, instead only show some figure. In addition, I think the edge of column of platform above the free surface does not need refinement.\
3. Please give the reason why the laminar simulation is carried out in OpenFOAM. In my opinion, the turbulence model should be used to capture the nonlinear wave loads.
4. Please specify the wave theory which is used to generate the wave. And the figure 6 should present the more results to make sure the wave is simulate accurately. Form the presented results, it is hard to estimate whether the wave amplitude decreases with time or not.
5. Why the platform move towards opposite directions at the beginning of simulations in Fig7(b). The difference of mean heave position can be removed in Fig7(c). Even though the difference of mean heave position is removed, the mean value of heave motion between OpenFAST and $OF^2$ is also very large. Under this condition, there is no wave, I think the mean value of heave should be very close. And the difference of heave force is quite small in Fig.8. Please check the results of heave motion again.
6. The legned of figure should be $OF^2$, not OF2.

---

## Author Comment (AC1)

**Answers to RC1 Comments**

March 31, 2023

Dear sir or madam,

Thank you very much for your kind comments on the manuscript. We have carefully considered all your appreciations and have revised the manuscript accordingly.

Please, find below our answers to each one of your comments. We hope that you agree with all our responses. We are looking forward to hearing from you.

Best regards,

Guillén Campaña-Alonso, Raquel Martín-San-Román, Beatriz Méndez-López, Pablo Benito-Cia, José Azcona-Armendáriz.

*1. In Table 1, what does "deterministic" mean in the load case description?*
We use the term "deterministic" to describe those load cases where the wind is both steady and uniform as those two presented in the paper, as it follows the nomenclature previously used in OC4 project [Robertson et al.(2014b)Robertson, Jonl Would you suggest a more suitable description for this type of load cases?

*2. On page 7 line 190, please add a comma before "and two flexible modes ..."*
Thank you very much for your appreciation and careful review. The manuscript has been corrected accordingly.

*3. Were wave forcing/damping zones included in the OpenFOAM/OF2 simulations?*
No, they were not included because the boundary conditions implemented in OpenFOAM do not need them. We assure that the boundary conditions are far enough from the platform ensuring it is not influenced by them.
At the outlet, the *shallowWaterAbsortion* boundary condition has been used for wave absorption. This boundary condition applies a zero gradient condition to the alpha field and to the velocity z-component (as it can be seen in OpenFOAM's documentation: https://www.openfoam.com/documentation/guides/latest/doc/guide-wave-modelling.html) while it sets to zero the other two velocity components.

*4. Please provide the total number of cells used in the OpenFOAM/OF2 simulations.*
As you suggest, the total number of elements will be added to the manuscript. The mesh is approximately composed by 2.3 million elements. As the floating platform is geometrically not too complex it can be meshed with a relatively small quantity of elements while preserving the geometry integrity. Furthermore, the refinement strategy

employed allowed us to reduce the total number of elements.

*5. In figure 5, it looks like the CFD mesh does not have a boundary-layer region next to the floater surface. What is the boundary condition on the platform surface? Free slip?*

Due to the fact that we are using morphing meshes, it is needed that the boundary condition applied on the platform surface accounts for its velocity. This is achieved by applying the *movingWallVelocity* boundary condition. This boundary condition would be equivalent to a no-slip one if the platform could not move.

In addition, the main objective of the manuscript is to demonstrate the feasibility of this new approach and how it could be used as an alternative tool to analyze and design floating offshore wind turbines. This exhaustive verification regarding the boundary layer influence will be performed in future studies. We appreciate your suggestion.

*6. In Figures 7 and 8, the agreement between the two approaches for LC 3.1\* appears acceptable. However, the results for LC 3.1 with waves show important differences. Apart from the opposite initial transient motion in surge, which the authors already pointed out and should be investigated further, there is also a large phase shift in the periodic steady-state motion. This is somewhat concerning because the same wave time series was used in both simulations. The authors are encouraged to investigate what is causing the phase shift.*

As you kindly pointed out, there was an issue with the wave time series initialization used in the OpenFAST simulation that induced a greater phase shift than the truly existing one. This has been rectified and the manuscript will be updated. Furthermore, it will also been added an analysis of the existing phase shift between the wave elevation and the heave force and the pitch moment. For OpenFAST, the phase shifts are consistent with the reference [Robertson et al.(2014a)Robertson, Jonkman, Masciola, Song, Goupee, Coulling, and Luan]. Nevertheless, comparing this results against $OF^2$ there exists a difference between these values collected in the following table.

| Phase shift | $OF^2$ | OpenFAST |
|---|---|---|
| Heave force | 173.56 deg | 181.84 deg |
| Pitch moment | -118.73 deg | -88.24 deg |

It is likely that these differences is due to the fact that OpenFAST computes the wave forces at the initial position of the platform whereas OpenFOAM/$OF^2$ computes them at the displaced position. In the $OF^2$ case, the displacement in surge is more than 5 meters causing this differences.

*7. Between lines 245 and 250 on page 11, the authors state that the mean displacements of the platform are similar between the two approaches; therefore, the mean drift load is consistently captured. In my opinion, the mean displacement observed in this case is primarily driven by the wind. In fact, Figure 8b shows the two methods giving rather different mean hydrodynamic force in surge. It's also surprising that OpenFAST appears to show a negative mean surge force, opposite of what's expected.*

As you suggested, we have analyzed the force in surge direction together with the surge displacement. We believe that the stationary states has not been fully reached and there are still some low frequency oscillations that difficult the analysis. Furthermore, considering that the mean force at this wave period should be possitive but very small. In consequence we have decided to run the analysis for 20 wave periods more in order to being capable of analysing the response in surge direction adequatelly. We are also considering to decrease the regular wave period so as to obtain higher mean drift forces. Due to lack of resources, we are not able to update the results in this submission, and we hope to do it in a few weeks.

*8. The OF2 results for pitch motion and moment appear to be slowly decaying, whereas the pure OpenFAST results do not. I'm curious if there is any potential explanation for this.*

As you pointed out, both the pitch displacement and moment are slowly decaying and the stationary state seems not to be reached. As we are going to run the simulation for 20 wave periods more, we will come back to this analysis in order to answer why this decay is happening. At the OF$^2$ simulation the free surface elevation has not been monitored in order to avoid a computation problem that sometimes occur when trying to sample a position within the platform geometry.

*9. Is the larger fluctuation of rotor speed and generator power observed in the pure OpenFAST results simply a consequence of the larger tower-top motion?*

Yes we agree with you that this is the main reason.

*10. While the hybrid approach presented in this paper is definitely more computationally efficient compared to full CFD simulations, the comparison of computing time needs to be treated with care because it heavily depends on the targeted level of numerical resolution and convergence. For example, Tran and Kim (2016) reported the use of a prism-layer mesh on the platform surface to help resolve the boundary layer. It can therefore be argued that the CFD simulations of Tran and Kim have a higher level of fidelity compared to the present study. The authors are encouraged to include discussions on these caveats in the comparison of CFD computing time.*

Yes, you are completely right and a more fair description of the compared simulations will be added to the manuscript. As you suggest, this is an interesting issue that should have been covered in the initial version of the mansucript.

**References**

[Robertson et al.(2014a)Robertson, Jonkman, Masciola, Song, Goupee, Coulling, and Luan] Robertson, A., Jonkman, J., Masciola, M., Song, H., Goupee, A., Coulling, A., and Luan, C.: Definition of the Semisubmersible Floating System for Phase II of OC4, Technical Report TP-5000-60601, NREL, 2014a.

[Robertson et al.(2014b)Robertson, Jonkman, Vorpahl, Wojciech, Frøyd, Chen, Azcona, Uzunoglu, Guedes Soares, Luan, Yutong, Robertson, A., Jonkman, J., Vorpahl, F., Wojciech, P.and Qvist, J., Frøyd, L., Chen, X., Azcona, J., Uzunoglu, E., Guedes Soares, C., Luan, C., Yutong, H., Pengcheng, F., Yde, A., Larsen, T., Nichols, J., Buils, R., Lei, L., Nygaard, T., Manolas, D., and He: Offshore Code Comparison Collaboration Continuation Within IEA Wind Task 30: Phase II Results Regarding a Floating Semisumersible Wind System, in: International Conference on Ocean, Offshore and Arctic Engineering, OMAE, 2014b.

---

## Author Comment (AC2)

**Answers to RC2 Comments**

March 31, 2023

Dear sir or madam,

Thank you very much for your kind comments on the manuscript. We have carefully considered all your appreciations and have revised the manuscript accordingly.

Please, find below our answers to each one of your comments. We hope that you agree with all our responses. We are looking forward to hearing from you.

Best regards,

Guillén Campaña-Alonso, Raquel Martín-San-Román, Beatriz Méndez-López, Pablo Benito-Cia, José Azcona-Armendáriz.

*As the hydrodynamics of platform is considered in OpenFOAM, and the motions of platform is transmitted to OpenFAST. Why the OpenFAST calculate the hydrodynamics again based on potential-flow and Morison equation? I think the coupling between OpenFOAM and OpenFAST should be explained in detail.*

Thank you for your comment. At the manuscript two different simulation approaches are employed: OpenFAST and OF$^2$. The OpenFAST approach, or OpenFAST-only model, is detailed between lines 215 and 230. The hydrodynamic model of this approach is based in a combination of potential-flow and Morison equation, using HydroDyn. On the other hand, the new approach described is OF$^2$. Under this approach, as explained in section 2, the aero-servo-elastic response of the wind turbine is simulated with OpenFAST, while the floating platform dynamics, hydrodynamics and fluid flow are simulated with OpenFOAM. Therefore, nor potential-flow nor Morison terms are included in $OF^2$, as the hydrodynamic response is resolved with this CFD model.

In the OF$^2$ approach, at the beginning of a time step, the hydrodynamic forces acting on the platform are computed with OpenFOAM and the influence of the wind turbine (loads at tower base) is also taken into account on the platform through the OpenFOAM restraint that we have developed contained in the libOF2.so dynamic library. To compute the wind turbine forces at the tower base, the displacement, velocity and acceleration of the floating platform reference point is imposed to the OpenFAST simulation. Once this forces are taken into account, the dynamics of the platform is solved and fluid flow is solved finishing the current iteration.

*1. Listing 1 can be given as an appendix.*

Thank you very much for your suggestion. We agree with you and we have modified the manuscript accordingly.

*2. I think it needs some words to introduce how the mesh around the platform's surface is refined, instead*

*only show some figure. In addition, I think the edge of column of platform above the free surface does not need refinement.*

As you suggest we have completed the mesh description including the mesh refinements employed and this will be added to the next version of the manuscript.

*3. Please give the reason why the laminar simulation is carried out in OpenFOAM. In my opinion, the turbulence model should be used to capture the nonlinear wave loads.*

According to [Wang et al.(2022)Wang, Robertson, Jonkman, Kim, Shen, Koop, Borràs Nadal, Shi, Zeng, Ransley, Brown, H where uncertainties from different turbulence modelling were studied, the uncertainty associated with the turbulence model is secondary to the discretization uncertainty. We assume, therefore, that analyzing the impact of the choice of turbulence model without performing a discretization analysis study is beyond the scope of the present study. Indeed, the main objective of the manuscript is to demonstrate the feasibility of the new approach. Therefore, these turbulence sensitivity analysis were not carried out.

*4. Please specify the wave theory which is used to generate the wave. And the figure 6 should present the more results to make sure the wave is simulate accurately. Form the presented results, it is hard to estimate whether the wave amplitude decreases with time or not*

The wave model employed to generate the wave is STOKES II according to the relationship between wave height, wave period and water depth. Figure 6 was included to demonstrate why there is not a oscillatory response of the FOWT at the start of the simulation. As you suggest, the complete wave elevation time series will be updated. Below you can see the complete wave elevation series.

[Figure]

Figure 1: Wave elevation evolution.

*5. Why the platform move towards opposite directions at the beginning of simulations in Fig7(b). The difference of mean heave position can be removed in Fig7(c). Even though the difference of mean heave position is removed, the mean value of heave motion between OpenFAST and OF 2 is also very large. Under this condition, there is no wave, I think the mean value of heave should be very close. And the difference of heave force is quite small in Fig.8. Please check the results of heave motion again.*

With regard to the surge displacement, we believe that OpenFAST usually struggles with initialization and that this could be the cause of this miss behaviour at the beginning of the simulation. We have also checked that the mooring system is configured equally in both approaches.

Regarding to heave behaviour, we appreciate your suggestion, we have checked again the heave displacement. Firstly, the OpenFAST model that has been used is the one defined by NREL at the DeepCWind definition document [Robertson et al.(2017)Robertson, Wendt, Jonkman, Popko, Dagher, Gueydon, Qvist, Vittori, Azcona, Uzunoglu, Soares, Secondly,for $OF^2$ simulations, we have employed the mass defined at the aforementioned document while the submerged volume is not user-defined but a result of the surface mesh employed. As collected in table 2, the mean heave displacement at the OpenFAST simulation is -0.03 m while at the $OF^2$ is -0,08m. It is true that a more refined surface mesh would lead to a smaller mean heave offset. Nevertheless, as the total draft of the platform is 30m, we have assumed that this deviation was negligible, as it represents a 0.3% of the platform draft.

*6. The legend of figure should be $OF^2$ , not OF2.*
Thank you very much for your correction. The manuscript has been corrected accordingly.

**References**

[Robertson et al.(2017)Robertson, Wendt, Jonkman, Popko, Dagher, Gueydon, Qvist, Vittori, Azcona, Uzunoglu, Soares, Harri
    Robertson, A. N., Wendt, F., Jonkman, J. M., Popko, W., Dagher, H., Gueydon, S., Qvist, J., Vittori, F., Azcona, J., Uzunoglu, E., Soares, C. G., Harries, R., Yde, A., Galinos, C., Hermans, K., De Vaal, J. B., Bozonnet, P., Bouy, L., Bayati, I., Bergua, R., Galvan, J., Mendikoa, I., Sanchez, C. B., Shin, H., Oh, S., Molins, C., and Debruyne, Y.: OC5 Project Phase II: Validation of Global Loads of the DeepCwind Floating Semisubmersible Wind Turbine, Energy Procedia, 137, 38–57, https://doi.org/10.1016/j.egypro.2017.10.333, 2017.

[Wang et al.(2022)Wang, Robertson, Jonkman, Kim, Shen, Koop, Borràs Nadal, Shi, Zeng, Ransley, Brown, Hann, Chandramo
    Wang, L., Robertson, A., Jonkman, J., Kim, J., Shen, Z.-R., Koop, A., Borràs Nadal, A., Shi, W., Zeng, X., Ransley, E., Brown, S., Hann, M., Chandramouli, P., Viré, A., Ramesh Reddy, L., Li, X., Xiao, Q., Méndez López, B., Campaña Alonso, G., Oh, S., Sarlak, H., Netzband, S., Jang, H., and Yu, K.: OC6 Phase Ia: CFD Simulations of the Free-Decay Motion of the DeepCwind Semisubmersible, Energies, 15, https://doi.org/10.3390/en15010389, 2022.

---

## Author Response (AR2)

**Answers to RC1 Comments**

June 22, 2023

Dear sir or madam,

Thank you very much for your comments on the revised manuscript. We find your suggestions enlightening.

Please, find below our answer to your new comments. We are looking forward to hearing from you.

Best regards,

Guillén Campaña-Alonso, Raquel Martín-San-Román, Beatriz Méndez-López, Pablo Benito-Cía, José Azcona-Armendáriz.

*5. In my opinion, if no prism-layer mesh region or wall function is used, it is better to use a free-slip condition on the platform wall. The authors should consider this in future investigations and perhaps mention this as an option in this paper.*

We have not consider the free-slip condition as we wanted to take into account the velocity of the platform and because of that we used the `movingWallVelocity` boundary condition. As you kindly suggest, we will consider the free-slip condition in further investigations.

*6. I agree that some phase shift between the OpenFAST and OF2 results is to be expected because OpenFAST evaluates the wave loads using the wave spectrum at the undisplaced platform position without considering any surge offset of the platform. The phase shift table in the author's reply also show the OpenFAST loads are leading the OF2 loads as expected. However, the phase shift between OF2 and OpenFAST is not the same for heave force and pitch moment. If the difference is indeed purely caused by the floater offset, one would expect the heave force and pitch moment to have the same phase shift. As a confirmation, perhaps the authors can calculate the expected phase shift between OpenFAST and OF2 based on the platform surge offset and the wavelength.*

As you suggest, we have computed the phase shift that would be expected if it would only be caused by the floater offset. For a mean surge displacement of 5 meters, the mean surge at the end of the simulation, the corresponding phase shift is $\sim 11°$. This suggest that there would be other effects that may affect to the phase shift. Nevertheless, we do not want to draw any conclusions in this respect due to the strong modulation of the waves (please, see next answer).

*With regard to the wave modulation.*

Firstly, in order to check the suitability of the spatial and temporal discretisation for the regular wave generation, free surface elevation was sampled at only one position. This sampling was used to verify that the wave had a proper amplitude and that the case configuration was able to keep the wave shape during the whole simulation time. The results show that only one sampling point is totally insufficient and a more thorough validation should have been carried out.

Therefore, we have checked the wave evolution along the wave channel without the structure and yes, we see a similar wave modulation as in Figure 9. We believe that this modulation comes from a misbehaviour of the boundary conditions.

In [Windt et al.(2019)Windt, Davidson, Schmitt, and Ringwood] a thorough study of numerical wave makers is carried out. In our work, we have used static boundary methods for both wave generation and absorption. At their research, Windt et al. found that the static boundary methods implemented by default in OpenFOAM are not as good as, for example, relaxation zone methods. This is due to their assumption of shallow water conditions for wave absorption.

At the manuscript, we have highlighted the misbehaviour of the boundary conditions employed that we believe they are the main source of error.

Because of this, we find that performing a detailed analysis of the phase shift between wave and load would draw misleading conclusions as the wave reflection could affect the phase of the computed loads.

Moreover, floating offshore wind turbines demand great time of simulation which has been found, [Larsen et al.(2019)Larsen, F to require suitable numerical schemes in order to keep the wave shape during the whole simulation. Having all of these issues in mind, we believe that wave generation and absorption is a critical aspect of FOWT simulations.

**References**

[Larsen et al.(2019)Larsen, Fuhrman, and Roenby] Larsen, B. E., Fuhrman, D. R., and Roenby, J.: Performance of interFoam on the simulation of progressive waves, Coastal Engineering Journal, 61, 380–400, https://doi.org/10.1080/21664250.2019.1609713, 2019.

[Windt et al.(2019)Windt, Davidson, Schmitt, and Ringwood] Windt, C., Davidson, J., Schmitt, P., and Ringwood, J. V.: On the assessment of numericalwave makers in CFD simulations, Journal of Marine Science and Engineering, 7, https://doi.org/10.3390/JMSE7020047, 2019.